# Dynamic Collaborations for the Development of Immune Checkpoint Blockade Agents

**DOI:** 10.3390/jpm11060460

**Published:** 2021-05-24

**Authors:** Arisa Djurian, Tomohiro Makino, Yeongjoo Lim, Shintaro Sengoku, Kota Kodama

**Affiliations:** 1Graduate School of Technology Management, Ritsumeikan University, Osaka 567-8570, Japan; gr0434hi@ed.ritsumei.ac.jp (A.D.); makino.tomohiro@gmail.com (T.M.); 2Faculty of Business Administration, Ritsumeikan University, Osaka 567-8570, Japan; lim40@fc.ritsumei.ac.jp; 3School of Environment and Society, Tokyo Institute of Technology, Tokyo 108-0023, Japan; sengoku.s.aa@m.titech.ac.jp; 4Center for Research and Education on Drug Discovery, The Graduate School of Pharmaceutical Sciences in Hokkaido University, Sapporo 060-0812, Japan

**Keywords:** cancer treatment, immune checkpoint blockade, PD-(L)1 inhibitors, interorganizational transaction, combination therapy

## Abstract

We studied the overview of drug discovery and development to understand the recent trends and potential success factors of interorganizational collaboration by reviewing 1204 transactions performed until 2019 for 107 anticancer drugs approved by the US Food and Drug Administration (FDA) from 1999 to 2018. Immune checkpoint blockade was found to be a significantly active area in interorganizational transactions, especially the number of alliances, compared with other mechanisms of action of small molecules and biologics for cancer treatment. Furthermore, the analysis of pembrolizumab and nivolumab showed that the number of approved indications for these two drugs has been rapidly expanding since their first approval in 2014. Examination of the acquisitions and alliances regarding pembrolizumab and nivolumab showed that many combination partners were developed by US-based biotechnology or start-up companies, the majority of which were biologics. These findings suggest that immune checkpoint blockade is a paradigm for cancer treatment, resulting in huge product sales and continuous indication expansion. Additionally, interorganizational collaboration, especially trial collaboration, is a strategic approach for the development of immune checkpoint blockade agents. The translation of these empirical practices to new drug candidates is expected for the research and development of innovative drugs in the future.

## 1. Introduction

Business in the pharmaceutical industry is unique in terms of product universality, where a company needs to research and develop each pipeline of drugs according to strict local regulations and requirements to obtain approval from the respective regulatory authority to launch their products. Therefore, drug research and development (R&D) is an extremely costly and time-consuming process [1,2,3,4]. In addition, business success in the pharmaceutical industry requires innovative products and expansion of product values to secure high pricing, grant of reimbursement and longer market exclusivities. Recently, R&D has become more complicated as modality becomes diverse by introducing new technologies and knowledge to address existing unmet medical needs in specific target populations. Thus, companies in the pharmaceutical industry have strived to enhance sustainability by adopting various strategic approaches, such as pursing globalized or region-oriented business models [5,6], selective therapeutic areas in R&D, optimizing R&D productivity [7], and open innovation, including external collaboration and interorganizational transactions [8,9,10,11]. In particular, the importance of external collaboration in R&D has been disseminated and many pharmaceutical companies have pursued interorganizational transactions and established open innovation platforms to acquire external knowledge and pipelines across organizations [8,10,11,12,13].

In the pharmaceutical industry, cancer therapeutics have a long history in the transformation of treatment options, including chemotherapy [14,15], targeted therapy [16], biologics [17], and combination therapy [18]. The recent hot spot is immuno-oncology [19,20], in which immune checkpoint blockade is a paradigm. Currently available immune checkpoint inhibitors are cytotoxic T lymphocyte-associated antigen 4 (CTLA4) inhibitors, programmed cell death protein 1 (PD-1) inhibitors, and programmed death-ligand 1 (PD-L1) inhibitors. The first entry into the market for immune checkpoint blockade was the anti-CTLA4 antibody ipilimumab (Bristol-Myers Squibb (BMS) (New York, NY, USA), and Medarex (Princeton, NJ, USA)), approved for metastatic melanoma in 2011 by the Food and Drug Administration (FDA) in the US. Subsequently, two anti-PD-1 antibodies, pembrolizumab (Merck) and nivolumab (BMS), were approved for metastatic melanoma in 2014 [21]. A number of PD-1 or PD-L1 inhibitors have been approved or are under development [22,23], thereby expanding the target tumor types [24] not only in the US but also in Europe, Japan, and other regions.

After the isolation of PD-1 by Dr. Honjo’s laboratory in Japan in 1992 [25], there was a long lag period until its first approval in 2014. The approval process involved extensive collaborations, supported by academia, biotechnology companies, pharmaceutical companies, biomarker companies, non-profit organizations, and regulatory agencies [20,26]. Chen and Han reviewed several important histories regarding anti-PD-(L)1 therapy for human cancer [26]. For example, the discovery of the PD pathway resulted from several collaborations by scientists, belonging to different laboratories, who identified and isolated at least five interacting molecules, PD-1 [25], PD-L1 (B7 homolog 1 [B7-H1]) [27,28], PD-L2 (B7-DC) [29,30], CD80 (B7-1) [31,32], and molecule family member b (RGMb) [33], and explored their functions and discovered mutual interactions [28,34,35,36,37]. Hoos suggested that collaboration among Medarex, BMS, a community organization called the Cancer Immunotherapy Consortium (CIC) of the Cancer Research Institute (New York, NY, USA), and CIC’s partners would generate breakthrough in the clinical evaluation of cancer immunotherapies by establishing new criteria and evaluation method. This led to the success of ipilimumab in clinical trials and eventually contributed to the acquisition of Medalex and responsible regulatory guidance by the FDA and the European Medicine Agency (EMA) (Amsterdam, The Netherlands). During the R&D of immune checkpoint blockade agents, there were important interorganizational transactions. For example, LifeArc (London, UK), which has a technology to generate a humanized clinical candidate, engineered pembrolizumab in collaboration with Organon in 2007 [38], and Merck acquired Organon later that year. Medarex, which was a biopharmaceutical company focused on the discovery, development, and potential commercialization of fully human antibody-based therapeutics and engineered ipilimumab and nivolumab, was acquired by BMS in 2009 [39]. Additionally, the development of combination therapy in this segment is quite active before and even after product launch [10,40,41] involving the collaboration of multiple companies. Given these previous successful practices, one of the effective approaches for sustainable business in the discovery and development of anticancer drugs may consider how a company that tries to generate a potential innovative drug can be a pioneer in terms of open innovation. Our interests were to understand whether immune checkpoint blockade is the active area in terms of interorganizational collaboration in anticancer drug development, and whether there are strategic collaborations with various external partners behind the recent successes of the indication and market expansion of immune checkpoint blockade agents. However, no previous study has investigated the trends in interorganizational collaborations for the R&D of immune checkpoint blockade that potentially led to successful drug development.

Therefore, the objective of this study was to understand the recent trends and success factors, based on various experiences of active transactions, for the development of immune checkpoint blockade agents. The definition of success in our study is a positive outcome derived from drug development or interorganizational transactions which enable a company to maintain sustainable growth, specifically drug approvals, new indication approvals, and market sales. We used the number of approved indications and market sales amount as proxies for success. We also investigated the important transactions in the discovery and development of immune checkpoint blockade agents, including extensive combination therapy development, especially pembrolizumab and nivolumab, which triggered the paradigm of standard of care in cancer therapeutics. In the present study, we investigated these from the angle of the mechanism of action (MOA), which defines how a drug or other substance produces an effect in the body; for example, how it affects a specific target in a cell, such as an enzyme, or a cell function, such as cell growth. Therefore, the efficacy and safety of drugs are highly dependent on the MOA [42]. Immune checkpoint blockade is the current major MOA in cancer therapy, and we expect that comparisons between immune checkpoint blockade and other MOAs of anticancer drugs are valuable for our research objectives. As an immune checkpoint blockade agent is a monoclonal antibody, that is, a biological anticancer drug, we precisely compared immune checkpoint blockade agents with other biologics. We also added small molecules for comparison, which are the most conventional therapeutic agents used in cancer treatment [14,15]. These small molecules were divided into two groups, namely kinase inhibitors (which are the major MOAs of small molecules) and other small molecules. Our goal was to uncover the potential key success factors of R&D for generating innovative and novel drugs to fulfill unmet medical needs in the future.

## 2. Materials and Methods

### 2.1. Samples and Data Sources

Sample data on cancer drugs were collected from the FDA’s New Molecular Entity (NME) list of approved small molecules and the New Biological Entity (NBE) list of approved biologics [24] from 1999 to 2018. Target drugs were determined based on CenterWatch’s list [43] of new cancer drug approvals, which was cross-referenced against the FDA’s NME and NBE lists. Using this approach, we selected 77 small molecules and 30 biologics as samples for this study.

The data source of approved indications for pembrolizumab and nivolumab was a package insert published by the FDA [24] from 2014 to 2019. The data source of product sales in pembrolizumab and nivolumab was Form 10-K from 2014 to 2019. The data source of company type and development phase of combination partners tested with pembrolizumab or nivolumab in interorganizational transactions was the Biomedtracker Deal Search [44].

### 2.2. Variables and Data Sources

Information on the number of transactions per product was collected from the Informa database’s Biomedtracker Deal Search [44]. We examined 1204 transactions related to the identified 107 cancer drugs approved by the FDA. The deal types defined by Informa included “acquisitions,” “alliances,” and “financing.” Each concept of “acquisitions,” “alliances,” and “financing.” are defined as follows:An acquisition is when one company purchases all or most of another company’s shares to gain control of that company.An alliance is an agreement between two or more companies regarding a pharmaceutical product, technology, service, etc.A financing involves a company raising money publicly or privately through the sale of equity, debt, or royalty monetization.

The deal characteristics defined by in the alliances consisted of “co-promotions,” “includes contract,” “includes equity,” “includes royalties or profit split information,” “intra-biotech deals,” “marketing-licensing,” “product or technology swaps,” “R+D and marketing-licensing,” “trial collaborations,” and “reverse licensing.” The development phase is derived from each company’s source information since the information on Biomedtracker Deal Search is obtained exclusively from publicly available sources such as company press releases, medical conference presentations, etc.

### 2.3. Statistical Analysis

SPSS Statistics 27 (IBM, Armonk, NY, USA) and Microsoft Excel 2016 (Microsoft, Redmond, WA, USA) were used for the statistical analysis. We performed multiple comparison Tukey–Kramer tests corresponding to 95% confidence intervals using a pre-set significance level (two-sided *p*-value of 0.05). The Tukey–Kramer test is the extension of Tukey’s test. Both Tukey’s test and the Tukey–Kramer test are a parametric multiple comparisons procedure and applies simultaneous to the set of all pairwise comparisons to find means that are significantly different from each other. We chose the Tukey–Kramer test which is used in the case of unequal samples sizes while Tukey’s test is used in the case of equal samples sizes. In the figures, the obtained *p*-values are presented as follows: *p* < 0.01 as **, *p* < 0.05 as *, and *p* < 0.1 as †.

## 3. Results

### 3.1. Interorganizational Transaction per Mechanism of Action

We initially investigated the trend of interorganizational transactions for drug discovery and development in the oncology area by reviewing 1193 transactions performed until July 2019. We performed a multiple comparison Tukey–Kramer test to compare the number of transactions among four categories of MOA: immune checkpoint blockade (which is the largest number of interorganizational transactions in biological anticancer drugs approved under Biologics License Application (BLA)), other biologics approved under BLA, kinase inhibitor (which is the largest number of interorganizational transactions in small molecule anticancer drugs under New Drug Application (NDA)), and other small molecules approved under NDA. The number of interorganizational transactions in immune checkpoint blockade was significantly higher than that in other MOAs (immune checkpoint blockade: other biologics: kinase inhibitors: other small molecules’ total = 44.00, SD 31.97: 11.30, SD 16.21: 6.95, SD 7.70: 9.26, SD 7.09, *p* < 0.05; acquisition = 3.29, SD 3.30: 1.17, SD 1.95: 0.68, SD 1.12: 1.46, SD 1.77, *p* < 0.05; alliances = 35.43, SD 24.67: 8.65, SD 12.63: 5.45, SD 5.94: 6.08, SD 5.40, *p* < 0.05; financing = 5.29, SD 4.89: 1.48, SD 2.25: 0.82, SD 1.66: 1.72, SD 2.25, *p* < 0.05; Figure 1). It has also been suggested that alliances are the major category of interorganizational transactions in immune checkpoint blockade.

We then explored the main purpose of transactions for alliances in immune checkpoint blockade. We classified the deal characteristics carefully. Some deals classified in deal types are classified two or more deal characteristics on Biomedtracker Deal Search, and we counted all deal characteristics in these cases. It was suggested that transactions for “alliance” were performed for trial collaborations most frequently, followed by R+D and marketing–licensing (Table 1).

### 3.2. Market Landscape and Interorganizational Collaborations for Combination Therapy Development of Top Two Immune Checkpoint Blockades: Pembrolizumab and Nivolumab

To understand the landscape of immune checkpoint blockade in the market, we examined the history of approved indications and annual product sales of two immune checkpoint blockades in the US, namely pembrolizumab and nivolumab, which triggered the paradigm of immuno-oncology and are the top two successful products in terms of the number of indications and product sales. Both pembrolizumab and nivolumab have achieved rapid indication expansion since they were initially approved in 2014 (Figure 2). Both products target various tumor types, such as solid and hematological tumors, including microsatellite instability-high and mismatch repair-deficient colorectal cancer. We also confirmed that product sales of both pembrolizumab and nivolumab have increased very rapidly, and the total annual sales of pembrolizumab surpassed that of nivolumab in 2018.

Next, we investigated the empirical landscape of combination therapy of pembrolizumab and nivolumab, as trial collaboration is the most common area in which active interorganizational transactions are performed in immune checkpoint blockade (Table 1), and most of them are related to combination therapy development. In this investigation, we selected only interorganizational transactions that directly influence the drug discovery and development of pembrolizumab and nivolumab, hereinafter referred to as effective interorganizational transactions. These included 67 transactions in alliance and financing for pembrolizumab until 2019 and 59 transactions in acquisition, alliance, and financing for nivolumab until 2019.

Figure 2 shows the annual number of effective interorganizational transactions for pembrolizumab and nivolumab, sorted into transactions for combination therapy development and others from 2014 to 2019. Throughout the period, the main purpose of transactions was combination therapy development for both pembrolizumab and nivolumab. Transactions for companion diagnostics (CoDx) development have been observed for both pembrolizumab and nivolumab. Ono Pharmaceutical and Dako AS contracted for CoDx development in 2015, BMS contracted with Enterome Bioscience SA for CoDx development in 2016 and invested in GeneCentric Diagnostics for translational biomarker research in 2017 for nivolumab, and Merck contracted with NanoString Technologies for CoDx development in 2016 for pembrolizumab. Interestingly, Merck lost two cases in 2017, where Merck paid $625 million upfront and global sales royalties to BMS owing to patent infringement to obtain non-exclusive rights and paid $19.5 million to PDL Biopharma owing to patent infringement to obtain non-exclusive rights.

Outside of the period from 2014 to 2019, one of the outstanding transactions was the contract between Ono and Medalex for joint R&D in 2005, in which Medalex obtained exclusive rights in North America. Of note, the deal search based on our algorithm in the database did not detect the acquisition of Medalex by BMS in 2009.

We also examined the features of combination partners, including developing companies, which were identified in effective interorganizational transactions for combination therapy development. As transactions in financing involve collaborations not directly related to combination therapy development, we only used transactions in acquisitions and alliances; there were 46 transactions in alliances for pembrolizumab and 41 transactions, including 3 acquisitions and 38 alliances, for nivolumab. Figure 3 shows the number of products distinguished by the type of company that develops combination partners tested with pembrolizumab or nivolumab, displayed in each development phase from preclinical to market. We classified developing companies into pharmaceutical companies or biotechnology company/start-up companies, and classified development phases into preclinical, phase 1, phase 1/2 or phase 2, phase 3, initial regulatory filing, or marketed after the initial regulatory approval. Some transaction information did not indicate the development phase of the combination partners and was classified as not applicable. It was observed that the main partners were biotechnology/start-up companies for both pembrolizumab and nivolumab; 8 pharmaceutical companies and 35 biotechnology/start-up companies for pembrolizumab and 8 pharmaceutical companies and 24 biotechnology/start-up companies for nivolumab. In addition, different trends were observed, as mainly combination partners under phase 1, phase 1/2, or phase 2 statuses developed by biotechnology companies or start-up companies were tested with pembrolizumab, whereas nivolumab was combined with combination partners under broader development phases from preclinical to marketed phases.

## 4. Discussion

We investigated the recent trends and success factors of interorganizational transactions in cancer therapeutics, especially immune checkpoint blockade. We also explored in depth transactions for the drug discovery and development of immune checkpoint blockade agents, including extensive combination therapy development, especially pembrolizumab and nivolumab. We collected data for each parameter from an open-source database and performed summary statistics and statistical analysis to study the trends of interorganizational transactions, combination therapies, especially with pembrolizumab and nivolumab, and the features of combination partners. Our findings suggest that immune checkpoint blockade is the most active area in terms of interorganizational collaboration in cancer drug development, where alliances related to combination therapy development are the center of practice. It was also suggested from a detailed investigation of transactions for pembrolizumab and nivolumab that different approaches were taken in indication expansion and phase of combination partners.

There were many more interorganizational transactions for drug discovery and development for immune checkpoint blockade than for other MOAs, and the main purpose of transactions was for trial collaboration, as shown in Figure 1 and Table 1, respectively. When we examined the details of interorganizational transactions for pembrolizumab and nivolumab, the majority of these were performed for combination therapy development. Based on these two observations, it was suggested that interorganizational collaborations are important in drug discovery and development of immune checkpoint blockade and alliances is the most effective means for activating combination therapy development which requires an in-licensing pipeline from another company or co-development with another company. These consequences may bring momentum to R&D activities in immune checkpoint blockade area by enabling partnering companies to achieve continuous regulatory authorization and lead business success in sales. In addition, indication expansions observed in Figure 2 may promote the potential combinations of immune checkpoint blockade with other agents for broader cancer types. Combination therapy is the key to expanding the use of pembrolizumab and nivolumab. This may be because combination therapy is expected to increase the number of responders who are non-responders with monotherapy, as previous studies argued that a combination strategy where various combination targets are tested with PD-(L)1 inhibitors is one of the most promising approaches to address treatment of these non-responders [21,40,45,46,47]. Although immune checkpoint blockade is a paradigm in cancer treatment, as shown by higher overall response rate, achievability of off treatment survival, and higher safety profile, there is clear challenge that only a fraction of cancer patients can derive clinical benefit from immune checkpoint blockade [21,47,48,49,50] thus much more research should be necessary for exploring further potential.

Both pembrolizumab and nivolumab are leading products in immune checkpoint blockade and both have common areas in drug discovery and development, including the same initial approval year for the same indication by the FDA. In addition, there are differences between pembrolizumab and nivolumab in terms of market landscape and interorganizational transactions. We anticipated that these differences could lead to successful drug discovery and development. Regarding the approval history and product sales transition of pembrolizumab and nivolumab, as shown in Figure 2, the indication expansion of nivolumab was initially more rapid than that of pembrolizumab from the first FDA approval in 2014 to 2017, indicating that the expansion of pembrolizumab increased from 2017 to 2019, whereas that for nivolumab has slowed down. This transition seems to link the transition of product sales, as initial product sales of nivolumab were higher than those of pembrolizumab until 2017, and this trend has been reversed since 2018. Both products are innovative and successful against various cancers. However, the difference in development strategies might have affected the result of indication expansion and product sales.

As shown in Figure 2, it was observed that the distribution of the effective transactions appears to present a “Gaussian” shape with the peak in 2017 while the sales trend keeps increasing even in 2019 [10]. Our previous study shows that that the observed distribution with the peak in 2017 is unique to pembrolizumab and nivolumab while other biologics and small molecules show different distribution of interorganizational transactions with peaks in different years. We anticipate that the reason for this is because newer immune checkpoint blockades and new drugs using new technologies such as gene therapy have been developed after the launch of pembrolizumab and nivolumab, and the interorganizational collaboration has shifted in combination with those newer agents. On the other hand, pembrolizumab and nivolumab still have solid positions with a large amount of evidence for various indications, which could be associated with an increase in sales as the previous research argued that the order of entry and strong efficacy are correlated with product sales [51,52]. In addition, we confirmed that several interorganizational transactions were performed for the R&D of CoDx and biomarkers. Although immune checkpoint blockade is a remarkably effective approach, previous researchers have argued that the overall response rate varies and is highly dependent on the microenvironment or genetic variations in patients with cancer [47,53]. Therefore, predictive biomarkers are very important [46,49,50,54] and PD-L1 expression is one of the most common biomarkers. However, it has been previously reported that a single biomarker is insufficient to predict clinical benefit or durability of the response to treatment in patients with cancer, and there are no reliable predictive biomarkers [21,50]. Therefore, our observations reflect the fact that further research in this area is necessary.

Regarding the observation of differences in the phase of combination partners between pembrolizumab and nivolumab, different strategies for partner selection can be expected. Although we did not investigate the correlation between strategic transactions and success in drug development and commercialization, the result shown in Figure 3 suggests that the selective approach of collaboration partners led to more successful indication expansion and increases in sales, as pembrolizumab was tested with many combination targets that were in phase 1, phase 1/2, or phase 2. It is also interesting to note that Merck’s transactions for payment owing to patent infringement seemed to be decisive, which enabled Merck to obtain exclusive rights to develop and sell pembrolizumab globally. In addition to interorganizational transactions, different approaches can be observed in clinical development, as in the case of nivolumab, which was ahead of pembrolizumab in clinical development entry; the duration from initial investigational new drug application to initial BLA submission was 100 months, with data from 886 subjects in six clinical trials, whereas it was 56 months with data from 1577 subjects in five clinical trials for pembrolizumab, which eventually enabled BLA approval earlier than that of nivolumab. These different approaches in drug development are referable when companies in the pharmaceutical industry develop pipeline strategies.

These considerations, based on our findings, can be translated into R&D strategies when new innovative pipelines are developed. In the future study, we will perform similar analysis focused on the MOA of additional combination therapy approved in recent years in order to clarify the findings in this study which were specific for immune checkpoint blockade.

## 5. Conclusions

The results of this study confirmed that active research and development related to immune checkpoint blockade is extraordinary in terms of continuous indication expansion and interorganizational transactions, especially for trial collaboration after the initial success of regulatory approval. In particular, trial collaboration for testing combinations with other cancer drugs was main deal characteristic specifically in the cases of pembrolizumab and nivolumab, which also achieved continuous indication expansion and increases in market sales. In addition, we found a remarkable variety of interorganizational collaborations, from the discovery of potential molecules, such as PD-1, to the post-launch of products.

However, there are limitations in our study. We performed database research using an open-source database and investigated only anticancer drugs, especially immune checkpoint blockade. In addition, detailed investigations were performed only for pembrolizumab and nivolumab. Further research is expected in the future to investigate whether the results from our study are applicable to other MOAs or modalities or whether the same kind of strategic collaboration can be seen in newer MOAs or modalities in the future. We expect the results of future research to address the remaining unmet medical needs and sustainability of the pharmaceutical industry.

## Figures and Tables

**Figure 1 jpm-11-00460-f001:**
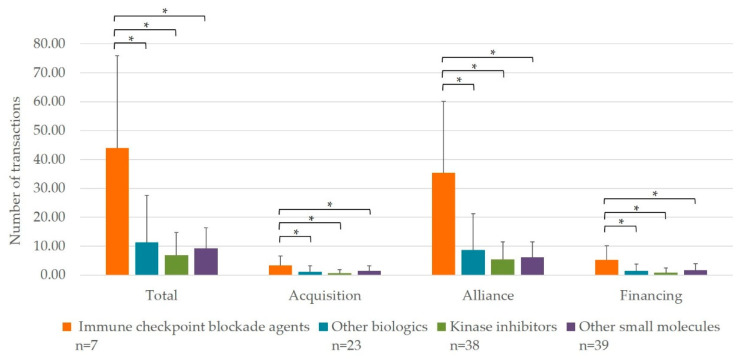
Multiple comparisons of the number of interorganizational transactions by the pairwise comparison procedure among immune checkpoint blockade agents, other biologics, kinase inhibitors and other small molecules. n = 77 (small molecules) and 30 (biologics). * *p* < 0.05.

**Figure 2 jpm-11-00460-f002:**
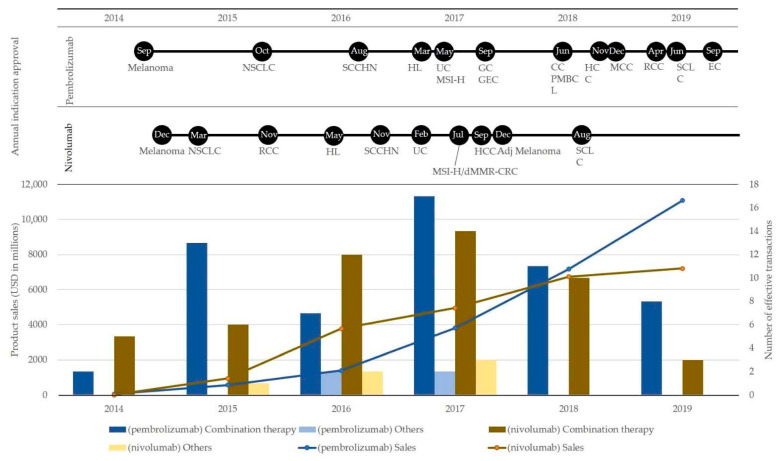
Annual approved indications, annual product sales (line graph), and annual number of effective interorganizational transactions (bar graph) of pembrolizumab and nivolumab from 2014 to 2019. NSCLC: non-small cell lung cancer, SCCHN: squamous cell carcinoma of the head and neck, HL: Hodgkin lymphoma, UC: urothelial carcinoma, MSI-H: microsatellite instability-high, GC: gastric cancer, GEC: gastroesophageal cancer, CC: cervical cancer, PMBCL: primary mediastinal large B-cell lymphoma, HCC: hepatocellular carcinoma, MCC: Merkel cell carcinoma, RCC: renal cell carcinoma, SCLC: small cell lung cancer, EC: endometrial carcinoma, BC: bladder cancer, TMB-H: tumor mutational burden-high, cSCC: cutaneous squamous cell carcinoma, dMMR: mismatch repair deficient, CRC: colorectal cancer, Adj Melanoma: adjuvant treatment of melanoma, ESCC: esophageal squamous cell carcinoma. Effective interorganizational transactions means interorganizational transactions that directly influenced the drug discovery and development of pembrolizumab and nivolumab. Combination therapy: interorganizational transactions for combination therapy development. Others: interorganizational transactions for other purpose.

**Figure 3 jpm-11-00460-f003:**
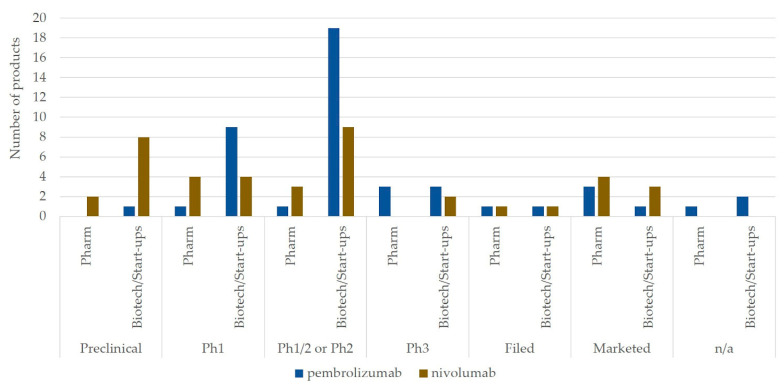
Number of products that were tested with pembrolizumab or nivolumab for combination therapy development. The number of products is displayed by the type of developing company as Pharm or Biotech/Start-ups. Pharm: pharmaceutical company, Biotech: biotechnology company, Start-ups: start-up company, Ph1: Phase 1, Ph1/2 or Ph2: phase 1/2 or phase 2, Ph3: phase 3, Filed: initial regulatory filing, Marketed: marketed after the initial regulatory approval, n/a: not applicable.

**Table 1 jpm-11-00460-t001:** The average number of interorganizational transactions for seven immune checkpoint blockade agents in each deal characteristic defined in the alliances.

	Average	SD
Co-promotion	3.571	2.225
Includes Contract	0.714	0.756
Includes Equity	2.143	2.116
Includes Royalty or Profit Split Information	9.000	5.916
Intra-biotech Deal	4.714	6.396
Marketing-licensing	0.714	1.496
Product or Technology Swap	0.429	0.535
R+D and Marketing-licensing	14.000	9.678
Trial Collaborations	24.429	18.636
Reverse Licensing	0.286	0.488

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
