# Peer review of "Dynamic Collaborations for the Development of Immune Checkpoint Blockade Agents"

_jpm, 2021, doi:10.3390/jpm11060460_

Round 1
Reviewer 1 Report
The meaning of Figures and Tables are hard to understand for clinicians. The authors should add the description of Figures and Tables legends for the easy understandings for readers. The Figures and Tables should be easily understood by clinicians without referring to main text for information about Figures and Tables.
Author Response
Re; Our responses to the reviewer’ comments
jpm-1220453: Dynamic Collaborations for the Development of Immune Checkpoint Blockade Agents
Thank you so much for your e-mail letter informing the decision of major revision and referees’ positive suggestions on our manuscript. We highly appreciate you to give us an opportunity to improve our paper.
I am sending herewith a revised version of our manuscript. Also our point-by-point responses to the referees’ comments are described in the following pages (2-5).
We believe that we revised as their comments and we hope our responses meet their expectations and intentions.
We like to sincerely express our gratitude for your consideration. I look forward to hearing from you.
Warm Regards,

Reviewer 2 Report
The manuscript “Dynamic collaborations for the development of immune checkpoint blockade agents” by Djurian et al. examined interorganizational collaborations for the immune checkpoint blockade such as pembrolizumab and nivolumab. The study in this manuscript showed the comparison of the number of interorganizational transactions (Figure 1) and approval history (Figure 2). Checkpoint inhibitors are well known as the most widely successful immunomodulators. Checkpoint inhibitors have been already integrated into many cancer types as a single treatment or combination in clinics. What is the main concept/purpose to analyze interorganizational transactions and approval history? The authors failed to emphasize the purpose and the rationale of their work and make a conclusion of each analysis.
Minor comments:
- Need to describe more details of each concept of “alliance”, “Acquisition”, and “Financing”. Why these are important in interorganizational transactions?
- Why is important that alliances are the major category of interorganizational transactions in immune checkpoint blockade? There are no clear conclusions.
- No Tukey-Kramer test description in Materials and Methods.
- In Table 1, Trial collaborations are the most common area interorganizational transactions. So, what’s the conclusion?
Author Response
Dear Reviewer
Re; Our responses to the reviewer’ comments
jpm-1220453: Dynamic Collaborations for the Development of Immune Checkpoint Blockade Agents
Thank you so much for your e-mail letter informing the decision of major revision and referees’ positive suggestions on our manuscript. We highly appreciate you to give us an opportunity to improve our paper.
I am sending herewith a revised version of our manuscript. Also our point-by-point responses to the referees’ comments are described in the following pages (2-5).
We believe that we revised as their comments and we hope our responses meet their expectations and intentions.
We like to sincerely express our gratitude for your consideration. I look forward to hearing from you.
Warm Regards,

Round 2
Reviewer 1 Report
The manuscript has been revised according to the reviewers’ comments.
Reviewer 2 Report
The authors have responded to the reviewer's concerns and comments and the manuscript has been much improved.